

# The prognostic model of low-grade glioma based on m6A-associated immune genes and functional study of FBXO4 in the tumor microenvironment

Yiling Zhang[1],[*], Na Luo[1],[*], Xiaoyu Li[2], Chuanfei Zeng[3], Xin Chen[1], Xiaohong Peng[1], Yuanyuan Zhang[4] and Guangyuan Hu[1]

[1] Department of Oncology, Tongji Hospital, Tongji Medical College, Huazhong University of Science and Technology, Wuhan, Hubei, China
[2] Department of Oncology, Hubei Cancer Hospital, Tongji Medical College, Huazhong University of Science and Technology, Wuhan, Hubei, China
[3] Renmin Hospital of Wuhan University, Wuhan, Hubei, China
[4] Department of Radiology, The First Affiliated Hospital, Zhejiang University School of Medicine, Hangzhou, Zhejiang, China
[*] These authors contributed equally to this work.

Corresponding authors
Yuanyuan Zhang,
z1731224497@163.com
Guangyuan Hu, h.g.y.121@163.com

## ABSTRACT

**Background:** m6A plays a dual role in regulating the expression of oncogenes and tumor suppressor genes, and is crucial in tumorigenesis and progression. The immune system is closely involved in tumorigenesis and development, playing a key role in tumor therapy and resistance. However, research on m6A-related immune markers in low-grade gliomas is still limited and requires further investigation.

**Methods:** All data was obtained from the Chinese Glioma Genome Atlas database and The Cancer Genome Atlas. The construction of the prognostic model and the online application of the dynamic nomogram relied on univariate Cox analysis, LASSO regression, and multivariate Cox analysis. Two different clustering analyses were performed on all samples, resulting in high, medium, and low expression groups of m6A regulatory and immune genes, followed by an analysis of the correlations between these scores. Finally, the biological role of FBXO4 in glioma cells was determined through quantitative reverse transcription polymerase chain reaction, cell proliferation assays, and cell migration experiments.

**Results:** The prognostic model for low-grade glioma demonstrated strong performance, with an AUC over 0.9 in the training group. In the internal validation group, AUC values ranged from 0.831 to 0.894, while in the external validation group, the AUC ranged from 0.623 to 0.813. Additionally, the online application of the dynamic nomogram allowed for relatively accurate predictions of LGG patients' survival time. Further analysis revealed that the high-expression groups of m6A regulatory genes and m6A-related immune genes exhibited higher levels of immune cells and stromal cells, lower tumor purity, and poorer survival rates. GSEA enrichment analysis suggested that these findings might be related to the activation of multiple signaling pathways. This may explain the lower survival rates observed in this group. Furthermore, the m6A score was significantly associated with moderate to high expression of immune genes and high expression of m6A regulatory genes, and it showed a positive correlation with most immune cell types. Finally, *in vitro*

experiments confirmed that silencing FBXO4 significantly inhibited proliferation and migration in glioma cell lines, further supporting the biological relevance of our model.

**Conclusion:** Based on multi-dimensional clustering analysis and experimental validation, the prognostic model developed in this study can effectively assess the prognosis of LGG patients and their relationship with the immune microenvironment. Furthermore, the correlation analysis between m6A scores and the tumor microenvironment provides a foundation for further exploration of the disease's pathophysiology. Additionally, we suggest that FBXO4 may serve as an important biomarker for the diagnosis and prognosis of LGG.

## INTRODUCTION

Gliomas, the most prevalent primary malignant tumors of the central nervous system (CNS), exhibit marked heterogeneity and invasiveness, with low-grade gliomas (LGGs, WHO grades I-II) representing 15–20% of all cases. Notably, aberrant m6A patterns are strongly associated with LGG prognosis (*Louis et al., 2021*; *Youssef & Miller, 2020*). Despite their relatively indolent growth, LGGs inevitably recur and transform into high-grade counterparts, with a 10-year progression rate exceeding 70% (*Perwein et al., 2021*). Current therapeutic strategies—maximal safe resection combined with adjuvant radiotherapy or chemotherapy—only modestly improve survival, underscoring the urgent need for biomarkers to refine risk stratification and guide immunotherapy (*Manoharan et al., 2020*). While molecular classifications (*e.g.*, IDH1/2 mutations) have improved prognostic prediction, they inadequately explain the immunosuppressive tumor microenvironment (TME) that drives LGG progression (*Kohanbash et al., 2017*; *Wijnenga et al., 2018*).

This study focuses on the role of m6A-related immune genes in low-grade gliomas (LGGs), driven by two primary motivations. First, although the functions of m6A modification in tumor immune regulation have been extensively studied, its specific mechanisms in LGGs remain poorly understood. Second, the high heterogeneity of the LGG immune microenvironment may be regulated by dynamic m6A modifications, yet systematic research to elucidate this potential association is still lacking. Therefore, this study aims to systematically analyze the expression patterns, prognostic significance, and regulatory networks of m6A-related immune genes in LGGs through integrated multi-omics approaches, with the goal of providing new theoretical foundations and potential molecular biomarkers for LGG immunotherapy strategies.

N6-methyladenosine (m6A), the most abundant internal RNA modification in eukaryotes, dynamically regulates RNA metabolism through "writers" (*e.g.*, METTL3), "erasers" (*e.g.*, ALKBH5), and "readers" (*e.g.*, YTHDF proteins) (*Boulias & Greer, 2023*). Dysregulated m6A modification is increasingly implicated in tumorigenesis and immune evasion. For instance, ALKBH5-mediated suppression of DNA repair genes (*e.g.*, CHK1,

RAD51) enhances glioblastoma stemness and radioresistance (*Lin et al., 2022b*), while YTHDF1/3 overexpression correlates with metastatic progression in breast cancer (*Anita et al., 2020*). Notably, aberrant m6A patterns are strongly associated with LGG prognosis (*Wu et al., 2024*), yet their role in shaping the immunosuppressive LGG TME—characterized by M2-polarized tumor-associated macrophages (TAMs) and IDH mutation-driven immune suppression remains unexplored (*Kohanbash et al., 2017*). This gap is critical, as emerging evidence links m6A to T-cell homeostasis and checkpoint inhibitor responses (*Chen et al., 2022*), suggesting its potential as a therapeutic target to reverse LGG immunosuppression.

The LGG TME is a complex ecosystem dominated by immunosuppressive elements. IDH-mutant tumor cells secrete the oncometabolite D-2-hydroxyglutarate (D-2HG), which inhibits T-cell activation and promotes TAM recruitment (*Bunse et al., 2018*). Concurrently, M2-polarized TAMs secrete anti-inflammatory cytokines (*e.g.*, TGF-β, IL-10), fostering an immune-excluded phenotype (*Hambardzumyan, Gutmann & Kettenmann, 2016*). Strikingly, m6A modification regulates both tumor-intrinsic and immune cell functions: METTL3-mediated m6A deposition in colorectal cancer suppresses CD8+ T-cell infiltration, while METTL3 inhibition synergizes with anti-PD-1 therapy to enhance tumor clearance (*Chen et al., 2022*). In T cells, m6A-dependent mRNA decay governs differentiation and exhaustion (*Li et al., 2017*). These findings suggest that m6A dysregulation in LGG may orchestrate immune-related gene (IRG) networks to sustain immunosuppression, yet systematic investigations are lacking. To address these gaps, we hypothesized that m6A-regulated IRGs critically modulate LGG immune evasion and clinical outcomes. Using multi-omics datasets (TCGA, CGGA), we constructed a prognostic IRG signature regulated by m6A modifiers through LASSO and Cox regression analyses. We further deciphered tumor microenvironment (TME) heterogeneity by correlating m6A-IRG clusters with immune infiltration, IDH status, and checkpoint expression. Additionally, we experimentally validated FBXO4, a top candidate gene, for its role in glioma cell proliferation and immune modulation. This study provides three key advances: mechanistically, we unveil NF-κB pathway activation as a central node linking m6A-IRGs to LGG immunosuppression; clinically, we develop a dynamic nomogram integrating m6A-IRG scores with IDH status for personalized prognosis prediction; and translationally, we identify FBXO4 as a druggable target to disrupt m6A-immune crosstalk in LGG.

By bridging m6A epigenetics with TME immunobiology, our work offers novel insights into LGG progression and paves the way for combinatorial therapeutic strategies targeting the m6A-immune axis.

# MATERIALS AND METHODS

## Data acquisition and differential analysis

Clinical information, mutation data, and agene expression dataset (HTSeq-FPKM) for LGG patients were obtained from the TCGA. To obtain an external validation set, we also downloaded datasets and clinical data for mRNAseq_693 and mRNAseq_325 from the CGGA website, and m6A-regulated genes were obtained from previous publications

(*Xu et al., 2020b*). Immune-related genes were extracted from the gene set enrichment analysis (*Subramanian et al., 2005*) (GSEA: www.gsea-msigdb.org) website. All the samples were obtained from tissues obtained from primary glioma patients *via* clinical surgical excision. All analyses were restricted to WHO grade I–II cases, and high-grade tumors (grades III–IV) were excluded during the preprocessing stage. All the data were preprocessed *via* the sva and Limma packages (*Ritchie et al., 2015*).

To control for multiple comparisons in differential gene expression analysis, we applied the Benjamini-Hochberg false discovery rate (FDR) correction method using the p.adjust() function in R (method = 'fdr'). This approach was chosen to balance the control of false positives while minimizing the loss of true positives, particularly given the large-scale nature of genomic data. Genes with an FDR-adjusted *P*-value < 0.05 and |log2 fold change| > 0.5 were considered statistically significant and included in subsequent analyses.

To obtain m6A-associated immune genes, we calculated correlations between immune genes and m6A-regulated genes and further selected the resulting results, which calculated an absolute correlation coefficient greater than 0.4 (*P*-value < 0.05). The Kaplan-Meier method and univariate Cox analysis were then used to perform survival analysis of the m6A-regulated genes, which in turn led to the desired hazard ratio (HR) value as well as its *P* value. A package (the Igraph package) with a correlation coefficient greater than 0.8 was used to plot the co expression network between these two genes. To obtain a heatmap that expresses the differences between the tumor and normal groups, we subsequently used the limma package, as others have done (*Luo et al., 2022*), to identify m6A-related immune genes (false discovery rate (FDR) < 0.05 and |log fold change (FC)| > 0.5).

## Prognostic modeling, mutation analysis and dynamic nomograms

We divided all the LGG samples from the TCGA database into two groups: the training group and the internal validation group, while also seeking the corresponding external validation group in the CGGA database. To obtain statistically significant m6A-related immune genes for prognosis, we analyzed the training group correlation data *via* multivariate Cox stepwise regression analysis, univariate Cox regression, and the LASSO algorithm. In addition, highly correlated genes were excluded in order to optimize the prognostic model. In order to distinguish between the high-risk and low-risk groups, we determined the median risk scores of the training group and divided them accordingly. Next, we plotted receiver operating characteristic (ROC) curves and survival curves to test the predictive power of the model.

After constructing the prognostic model, we visualized and analyzed the data from the maftools package (*Mayakonda et al., 2018*) by producing mutation waterfall plots. We next analyzed the effect of mutational load on survival in both groups. To make our prognostic model more clinically useful, we downloaded the DynNom software package (*Jalali et al., 2019*) from the web and created a corresponding web-based dynamic histogram application, which helped us to determine the prognosis of our patients more accurately.

## ssGSEA analysis and cluster typing

The samples from both TCGA and CGGA databases were analyzed *via* ssGSEA *via* the GSVA software package to determine the number of immune cells present in each sample. To distinguish between high and low expression of m6A-regulated genes, we performed a cluster analysis of the genes obtained separately *via* the ConsensusClusterPlus package (*Wilkerson & Hayes, 2010*). We also plotted the violin, heatmap, and survival curve of the tumor microenvironment for the above three groups to determine the correlation between m6A and immunity.

We then performed a cluster analysis of m6A-associated immune genes affecting prognosis ($P < 0.05$, top 50), *via* the same method, to obtain groups with high, medium, and low expression of immune genes. We then also plotted violin plots, heatmaps, and survival curves of the tumor microenvironment capable of expressing correlations between m6A-related immune genes and immunity. Finally, we mapped multigene enrichment curves by using the org.Hs.eg.db R package. Our results demonstrate significant enrichment of five Gene Ontology (GO) terms and pathways from the Kyoto Encyclopedia of Genes and Genomes (KEGG) in the high expression group of immune genes compared to the low expression group ($P < 0.05$).

## m6A score calculation and sankey plot

We used the limma package to locate DEGs in the low-, medium-, and high-expression groups of m6A-regulated genes and then used their intersections *via* the VennDiagram package to obtain the final list of DEGs. The prognosis-related genes were then screened again *via* the one-way Cox method ($P < 0.05$). After obtaining information on the expression of prognosis-related genes, we calculated the m6A score for each sample *via* PCA. The resulting m6A score is then sorted into high and low groups *via* the surv_cutpoint function, and the corresponding survival curves are plotted.

We plotted Sankey diagrams to visualize the relationships among immune gene type, immune gene type, m6A score, and risk score. Correlation matrices for m6A score and immune cells were also generated *via* the corrplot package. Finally, we determined the correlation between them.

## Comprehensive analysis of model genes

Furthermore, we also collated correlations between individual model genes and parameters such as stem cell indices, immune subtypes, the tumor microenvironment, and clinical stage. In addition, to validate the prognostic role of the model genes, the Gene Expression Profile Interaction Analysis (GEPIA) database was selected for analysis. Finally, to validate the accuracy of the prognostic model, we used a 5-year ROC curve, which enables the comparison of risk values, tumor inflammatory characteristic (TIS) scores, and TIDE scores and assesses the prognostic predictive efficacy between them.

## Cell lines and siRNA transfection

The human astrocyte cell line HA1800 was obtained from the Shanghai Institutes for Biological Sciences (Chinese Academy of Sciences, Shanghai, China). The glioma cell lines

U87-MG (HTB-14™), and U118-MG (HTB-15™) were obtained from the American Type Culture Collection (ATCC). U251-MG were purchased from Boster Biological Technology, Ltd. (Wuhan, China). The culture temperature was 37 °C, and the concentration of $CO_2$ used was 5%. Our culture conditions included Dulbecco's modified Eagle medium (DMEM) (HyClone, Logan, UT, USA) supplemented with 10% fetal bovine serum (Gibco, Waltham, MA, USA). Our siRNA (100 nM) was purchased from RIBO Biotechnology, and after 24–48 h of incubation, the cells were transfected with the siRNA *via* an *In vitro* RNATMtransfection reagent (InvivoGen Biotechnology, Hong Kong, China).

## qRT-PCR

Total RNA was extracted from the HA1800, U87, U118, and U251 cell lines, as well as siRNA-transfected U87 and U251 cell lines (including their respective blank controls), using TRIzol reagent (Invitrogen, Waltham, MA, USA). The RNA concentration and purity were measured using a spectrophotometer. Then, 1 µg o Next, quantitative PCR (qPCR) was performed using gene-specific primers (listed below) and SYBR Green reagent in a 96-well plate. The qPCR conditions were as follows: initial denaturation at 95 °C for 30 s, followed by 40 cycles consisting of denaturation at 95 °C for 10 s and extension at 60 °C for 30 s. β-actin was used as the internal reference gene, and relative expression levels were calculated using the ΔΔCt method. RNA was reverse-transcribed into cDNA. The primers used for manipulation were as follows:

    FBXO4-Forward: GTCTTCGGGAAGACCATTGTTGG;
    FBXO4-Reverse: GAGTTTCAGCCTCTGTATCCTGG;
    β-actin-Forward: GACCACAGTCCATGCCATCA;
    β-actin-Reverse: GTCAAAGGTGGAGGAGTGGG.

## Cell proliferation

CCK-8 can be used to detect the proliferation of cells. We cultured U87 and U251 cells transfected with the FBXO4 siRNA for 48 h in a 96-well plate. In 96-well plates, 2,000 cells were cultured in each well with 100 µl of DMEM containing 10% FBS. We examined the value-added intensity of the treated cells every 24 h and obtained data for 5 days. In accordance with the instructions, we added equal amounts of CCK8 reagent (Yeasen, Shanghai, China) to each well of the plate and, after 1 h, absorbance values were obtained at 450 nm *via* a microplate reader (BioTek, Winooski, VT, USA).

## Migration assay

Transwell assays were used to examine cell migration. For the transwell assay, 40,000 cells were inoculated into the upper chamber of a Matrigel transwell plate (BD Biosciences, Milpitas, CA, USA). After 24 h of incubation, the inserts were cleaned with 1× phosphate buffer solution, and the cells were fixed in 95% ethanol and finally stained with 1% crystal violet for 20 min at room temperature. Finally, we photographed the perforated cells in their microscopic state and then used ImageJ to calculate the number of cells that crossed the membrane.

## Statistical analyses

We used R software (version 4.0.3) for routine statistical analyses. Some of the analysis data were also obtained from publicly available information bases and special software. Graph Pad (version 8.0.1) was also used for some of the graphs. The construction of the prognostic models relied on univariate Cox regression, multifactor Cox stepwise regression, and the LASSO algorithm. $P < 0.05$ was considered statistically significant.

# RESULTS

## Prognostic and immune co-expression of m6A genes

The clinical information we needed, derived from the TCGA database ($N = 512$), CGGA693 database ($N = 443$) and CGGA325 database ($N = 186$), has been collated in Table 1, and the results of survival analyses with respect to the m6A regulatory genes have been uploaded in File S1. Among the 22 m6A regulatory genes, except for METTL3, HNRNPC, LRPPRC, IGFBP1, and RBMX, all m6A regulatory genes were prognostic factors for LGG. The m6A prognostic network map clearly revealed a positive correlation between most m6A-regulated genes (Fig. 1A), with a few negative correlations indicated by blue lines. It also presents the $P$-values from the univariate Cox regression analysis, represented by the size of the dots, as well as whether each gene is a risk factor or a favorable factor, indicated by the color of the right half of the dots. In the co-expression network of immune genes and m6A-regulated genes, when the absolute value of the correlation coefficient is greater than 0.8, YTHDF2 exhibits the strongest correlation with most immune genes (Fig. 1B). Then, a correlation analysis was performed between m6A genes and immune genes, followed by batch correction of TCGA and CGGA samples, resulting in 7,757 m6A-associated immune genes. Subsequently, differential analysis was conducted between the normal group ($N = 5$) and the tumor group ($N = 529$), identifying 7,654 differentially expressed m6A-associated immune genes (FDR < 0.05 and |logFC| > 0.5). The differential expression results of these m6A-associated immune genes are visualized in a heatmap (Fig. S5).

## Identification and validation of prognostic m6A-related immune genes: construction of a predictive model

We combined the differential expression profiles of m6A-related immune genes with survival time and survival status, and used the coxph function to evaluate the impact of these genes on survival time, identifying a total of 3,229 m6A-related immune genes associated with prognosis ($P < 0.05$). We then performed LASSO regression on the 3,229 genes, using penalty terms to shrink some coefficients to zero, narrowing the candidate genes down to nine (Figs. 2A, 2B). These nine genes were subsequently incorporated into a multivariate Cox regression model to assess their independent effects on survival. Ultimately, we identified four genes with a significant impact on prognosis. These four genes form the primary features of our prognostic model. The risk score formula for each sample is as follows: risk score = $0.157CRNDE + 0.160CUL7 + 0.505FBXO4 + 0.265TNFAIP6$ (Fig. 2C, Table 2). Specific information on the training group, the internal validation group, and the external validation group is included in Files S2–S5, respectively.

**Table 1  Clinical characteristics of LGG patients in the TCGA, CGGA693 and CGGA325 databases.**

| Characteristics | TCGA Total patients (N = 512) | | CGGA693 Total patients (N = 443) | | CGGA325 Total patients (N = 186) | |
|---|---|---|---|---|---|---|
| | No | % | No | % | No | % |
| **Age (y)** | | | | | | |
| <60 years | 443 | 86.52 | 426 | 96.16 | 174 | 93.55 |
| ≥60 years | 69 | 13.48 | 16 | 3.61 | 12 | 6.45 |
| NA | | | 1 | 0.23 | 0 | 0 |
| **Gender** | | | | | | |
| Female | 228 | 44.53 | 192 | 43.34 | 71 | 38.17 |
| Male | 284 | 55.47 | 251 | 56.66 | 115 | 61.83 |
| **Grade** | | | | | | |
| I | 0 | 0 | 0 | 0 | 0 | 0 |
| II | 247 | 48.24 | 188 | 42.44 | 103 | 55.38 |
| III | 265 | 51.76 | 255 | 57.56 | 79 | 42.47 |
| IV | 0 | 0 | 0 | 0 | 0 | 0 |
| NA | 0 | 0 | 0 | 0 | 4 | 2.15 |
| **Survival status** | | | | | | |
| Alive | 403 | 78.71 | 247 | 55.76 | 87 | 46.77 |
| Dead | 109 | 21.29 | 157 | 35.44 | 93 | 50 |
| NA | | | 39 | 8.8 | 6 | 3.23 |
| **Histology** | | | | | | |
| A | – | – | 129 | 29.12 | 67 | 36.02 |
| O | – | – | 81 | 18.28 | 38 | 20.43 |
| OA | – | – | 233 | 52.6 | 77 | 41.4 |
| NA | | | 0 | 0 | 4 | 2.15 |
| **IDH1_mutation_status** | – | – | | | | |
| Wildtype | – | – | 96 | 21.67 | 51 | 27.42 |
| Mutation | – | – | 306 | 69.07 | 134 | 72.04 |
| NA | – | – | 41 | 9.26 | 1 | 0.54 |
| **1p19q_codeletion_status** | | | | | | |
| Non-codel | – | – | 273 | 61.63 | 121 | 65.05 |
| Codel | – | – | 131 | 29.57 | 60 | 32.26 |
| NA | – | – | 39 | 8.8 | 5 | 2.69 |
| **Radiation therapy** | | | | | | |
| Yes | – | – | 316 | 71.33 | 152 | 81.72 |
| No | – | – | 87 | 19.64 | 25 | 13.44 |
| NA | – | – | 40 | 9.03 | 9 | 4.84 |
| **Chemotherapy** | | | | | | |
| Yes | – | – | 266 | 60.05 | 85 | 45.7 |
| No | – | – | 125 | 28.22 | 84 | 45.16 |
| NA | – | – | 52 | 11.73 | 17 | 9.14 |

**Note:**

A, astrocytoma; NA, not available; O, oligodendroglioma; OA, oligoastrocytoma; IDH1, isocitrate dehydrogenase 1.
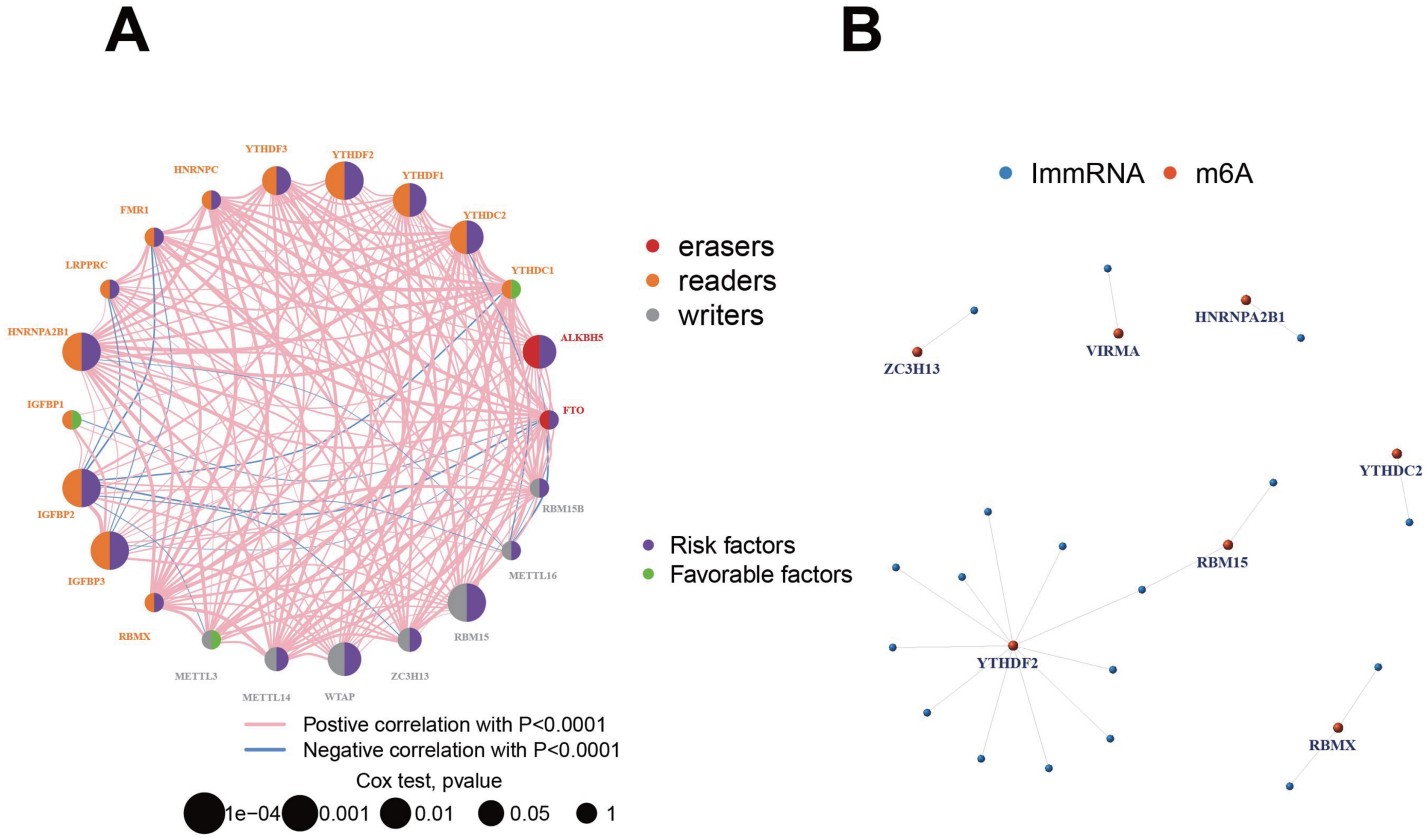

**Figure 1 Visualization of m6A regulatory and immune genes.** (A) Prognostic network diagram of m6A. Nodes represent m6A regulatory genes, with the left semicircle color indicating different types of m6A regulators, where red, orange, and gray represent erasers, readers, and writers, respectively. The right semicircle color indicates whether the m6A regulatory gene is a high-risk gene, with blue representing high-risk. The size of the nodes reflects the relationship between the gene and survival, with larger nodes suggesting a higher likelihood of being associated with prognosis. Red lines indicate positive co-expression relationships, and blue lines indicate negative co-expression relationships. (B) Co-expression network of m6A regulatory genes and immune genes. Red nodes represent m6A regulatory genes, blue nodes represent immune genes, and connecting lines represent co-expression relationships.

We calculated the median risk score from the training group and used this value to classify the samples in the training, internal validation, and two external validation groups into low-risk and high-risk groups. Survival analysis showed that, although survival varied across groups, the survival rate in the low-risk group was significantly higher than that in the high-risk group. We also performed AUC testing under the ROC curve in all four groups to validate the predictive efficacy of the model (Figs. 2D–2G). The results revealed an AUC value greater than 0.9 in the training group, an AUC ranging from 0.831 to 0.894 in the internal validation group, and an AUC between 0.623 and 0.813 in the external validation groups. These results suggest that our prognostic model has high predictive power in LGG.

## The prognostic value of m6A-related immune genes and mutational load

We systematically evaluated whether riskScore has independent prognostic predictive ability for patients through univariate and multivariate Cox regression analyses. In the

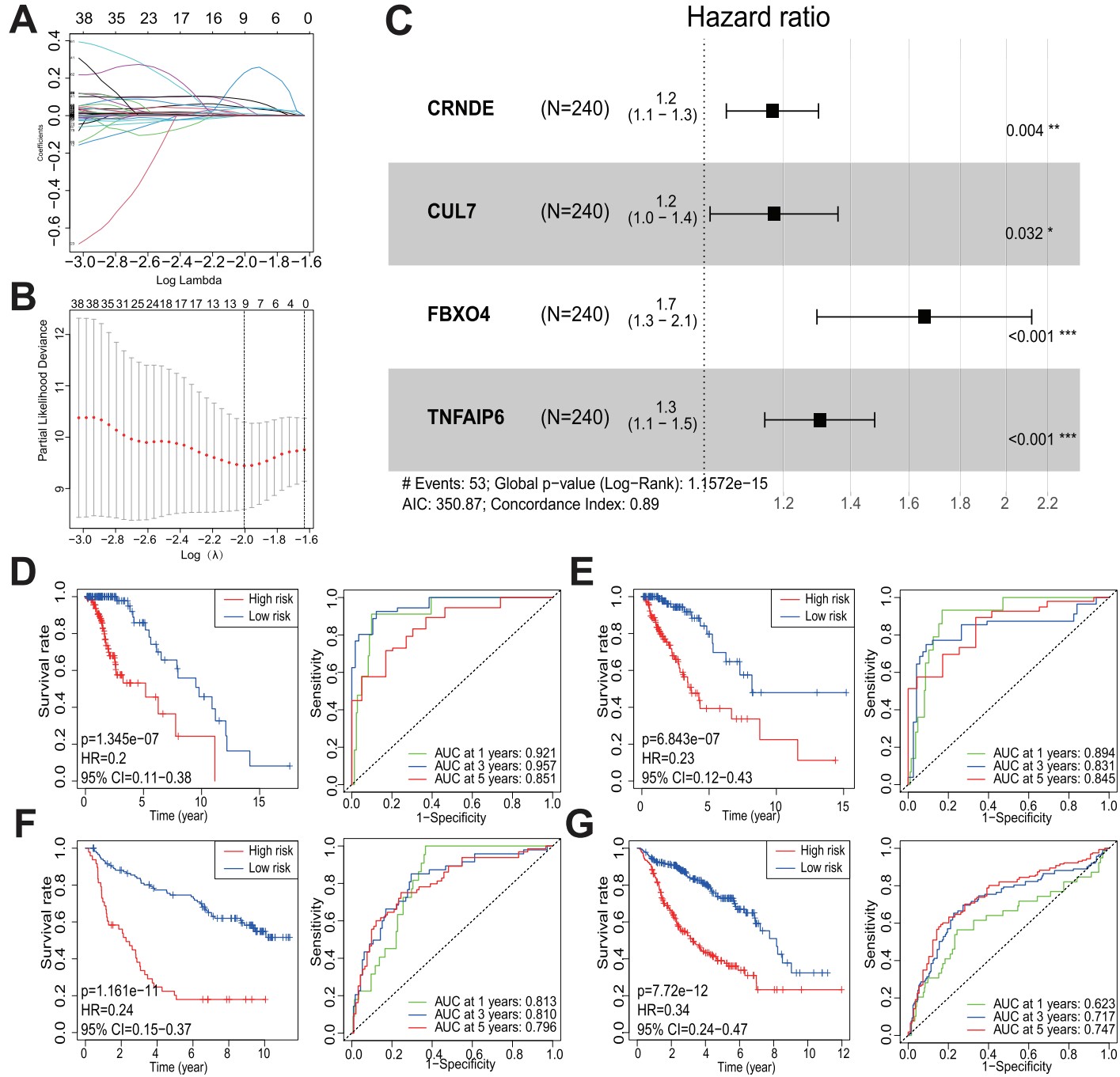

**Figure 2 Establishment and evaluation of the clinical prognostic model.** (A) LASSO coefficient plot. The 3,229 m6A-related immune genes significantly associated with prognosis were identified using the Cox regression model. (B) λ selection plot. In the training group, the optimal log lambda value was selected in the LASSO model, narrowing down the candidate genes to nine. (C) Forest plot. Based on the multivariate Cox regression model, the forest plot shows the hazard ratios (HR) and 95% confidence intervals for all model genes. (D) Evaluation of the clinical prognostic model in the training group using survival curves and ROC curves. (E) Evaluation of the clinical prognostic model in the internal validation group using survival curves and ROC curves. (F) Evaluation of the clinical prognostic model in the external validation group (CGGA mRNAseq 325 dataset) using survival curves and ROC curves. (G) Evaluation of the clinical prognostic model in the external validation group (CGGA mRNAseq_693 dataset) using survival curves and ROC curves.

**Table 2 Multivariate Cox regression analysis results of model genes.**

| id | coef | HR | HR.95L | HR.95H | P value |
|---|---|---|---|---|---|
| CRNDE | 0.156797184 | 1.169758344 | 1.052374174 | 1.300235806 | 0.003659506 |
| CUL7 | 0.160436226 | 1.174022898 | 1.013751852 | 1.359632304 | 0.032165755 |
| FBXO4 | 0.505422079 | 1.657685047 | 1.29557519 | 2.121003657 | 5.84E−05 |
| TNFAIP6 | 0.265369042 | 1.303912086 | 1.149370074 | 1.479233509 | 3.74E−05 |

multivariate analysis, we further validated whether riskScore could independently predict the prognosis of patients after controlling for clinical variables such as age, sex, and grade. The results showed that riskScore had a significant $P$-value and reasonable HR in both the univariate Cox regression and multivariate analysis (Figs. 3A, 3B), indicating that riskScore can serve as an independent prognostic factor and help predict the survival of LGG patients. Mutation waterfall plots revealed the genes with the highest mutation frequencies, which were IDH1, TPP3, and ATRX (Figs. 3C, 3D). Further analysis revealed that mutational load had a significant effect on survival in both groups (Fig. 3E). To make it easier to calculate patient survival, we have designed a web-based dynamic histogram application (available at https://u20131050.shinyapps.io/LGG-m6A_ImmRNA-Dynamic_ nomogram/) in which the expression levels of the model genes can be entered directly to obtain the desired results.

## Sample clustering based on m6A regulatory genes and tumor TME scoring

In the clustering analysis, we determined the optimal number of clusters to be three based on the cumulative distribution function (CDF) plot, as the curve for K = 3 was tighter and smoother compared to K = 2. Based on the expression levels of m6A-regulating genes in TCGA and CGGA samples, the samples were divided into high, medium, and low expression groups. The specific classification and expression levels of each gene are shown in the heatmap (Fig. 4A). Survival analysis revealed a negative correlation between the expression of m6A-regulating genes and survival rate (Fig. 4B). Although no statistical difference was observed between the medium and low expression groups, significant statistical differences were found between the medium-high and high-low expression groups ($P < 0.05$). Through the analysis of heatmaps and violin plots of the tumor microenvironment, we observed that the high-expression group exhibited higher immune cell and stromal cell content, leading to enhanced immune activity and lower tumor purity. In contrast, the medium- and low-expression groups showed the opposite trend (Figs. 4C–4G).

## Sample clustering based on prognostic m6A-related immune genes and tumor TME scoring

We grouped the TCGA samples into high, medium, and low expression categories based on the expression levels of prognostic m6A-related immune genes. The specific classification of samples can be visualized in the heatmap (Fig. 5A). Survival analysis

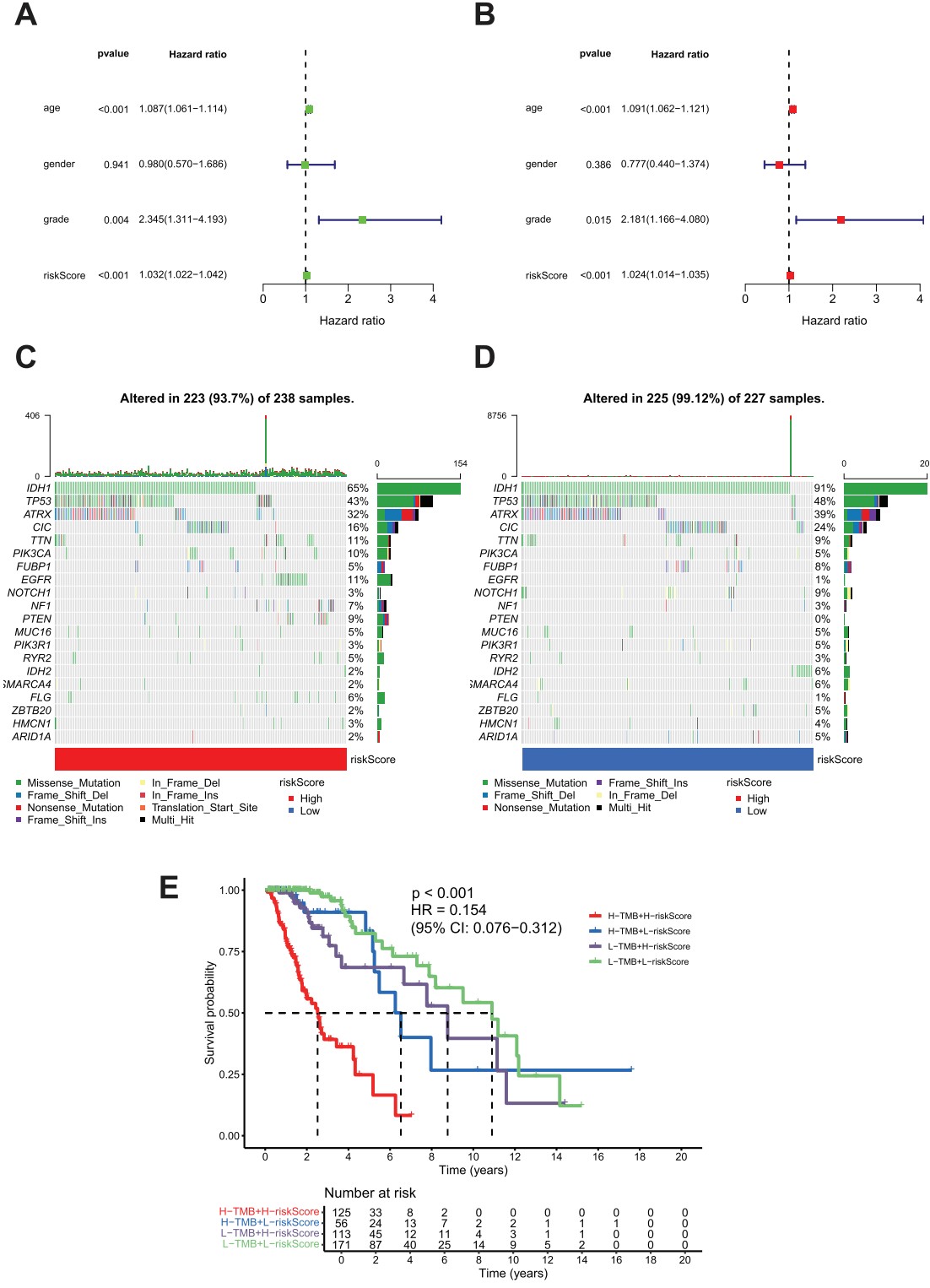

**Figure 3 Evaluation of the independent prognostic value of riskScore and mutation analysis.** (A) Univariate regression analysis. (B) Multivariate regression analysis. In both univariate and multivariate independent prognostic analyses, the *P*-value for riskScore was less than 0.05, indicating that riskScore is a potential independent prognostic factor. (C) Mutation waterfall plot for the high-risk group. Mutation frequencies are displayed on the right. (D) Mutation waterfall plot for the low-risk group. Mutation frequencies are displayed on the right. (E) Survival analysis combining mutation load and risk values. The hazard ratio (HR) and 95% confidence interval are shown.

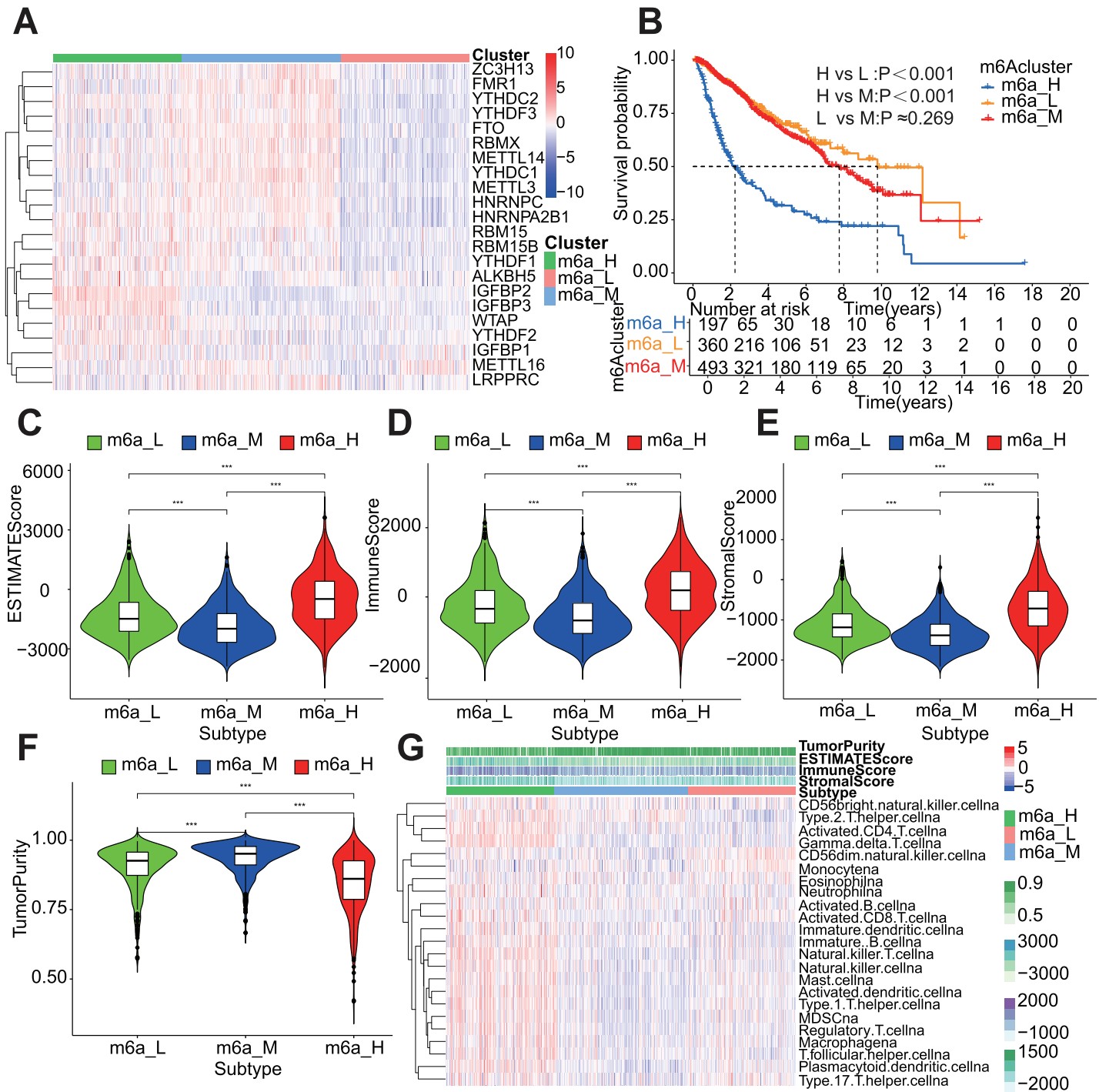

**Figure 4 Sample clustering based on m6A regulatory genes and tumor microenvironment (TME) scoring.** (A) Clustering heatmap. Displays the classification and expression levels of each gene. (B) Survival curves for the three different expression groups. (C) Violin plot of ESTIMATES scores across different groups. (D) Violin plot of immune scores across different groups. (E) Violin plot of stromal scores across different groups. (F) Violin plot of tumor purity across different groups. (G) Heatmap. The x-axis represents sample names, and the y-axis represents different immune cell types and their corresponding TME scores. In the violin plots, ***$P < 0.001$.

revealed that the low immune gene expression group had a significantly higher survival rate than the other two groups (Fig. 5B). Further analysis showed that in the high-expression group, the levels of immune and stromal cells were higher, indicating increased immune activity and lower tumor purity. In contrast, the medium and low-expression groups showed lower levels of immune and stromal cell content, with higher tumor purity (Figs. 5C–5G).

In addition, GSEA enrichment analysis showed that the high-expression group of m6A-related immune genes was significantly enriched in pathways such as the negative regulation of alpha-beta T cell activation, negative regulation of CD4-positive alpha-beta T cell activation, negative regulation of macrophage migration, negative regulation of T cell receptor signaling pathway, and T cell tolerance induction in GO enrichment analysis (Fig. 6A, only the five most significant pathways are listed). In the KEGG enrichment analysis, this group was significantly enriched in the IL-17 signaling pathway, JAK-STAT signaling pathway, NF-κB signaling pathway, TNF signaling pathway, and Toll-like receptor signaling pathway (Fig. 6B, only the five most significant pathways are listed). Taken together, these results reveal an interesting phenomenon: in patients with high expression of prognostic m6A-related immune genes, multiple immune-suppressive pathways are activated, which may be one of the reasons for the lower survival rate observed in this group.

## Prognostic significance of m6A scores and their association with immune cell infiltration in LGG

All information about the sample m6A scores can be found in File S6. As shown by the survival curves, the survival of the high m6A score group was significantly worse that of than the low m6A score group (Fig. 7A). The Sankey plots and box line plots revealed important correlations: the high m6A score group corresponded to the moderate immune gene expression group, the high immune gene expression group, and the high m6A regulatory gene expression group (Figs. 7B–7D). In addition, the correlation matrix clearly revealed significant positive correlations between m6A score and most immune cell types, although negative correlations were observed with CD56dim natural killer cells and monocytes. The above results revealed that the m6A score was strongly positively correlated with immune infiltration (Fig. 7E).

## Association of model genes with clinical stage, immune subtype, tumor microenvironment, and stem cell index

The associations between the selected model genes and clinical stage, immune subtype, tumor microenvironment, and stem cell index were compared with the help of our constructed prognostic model. The results of the present study revealed that the immune subtype and clinical stage were significantly correlated with the selected model genes (Fig. S1). Additionally, stratification of both the tumor microenvironment and stem cell indices yielded significant correlations with model genes (Fig. S2). From the data extracted from the GEPIA database, we can conclude that all model genes are high-risk prognostic

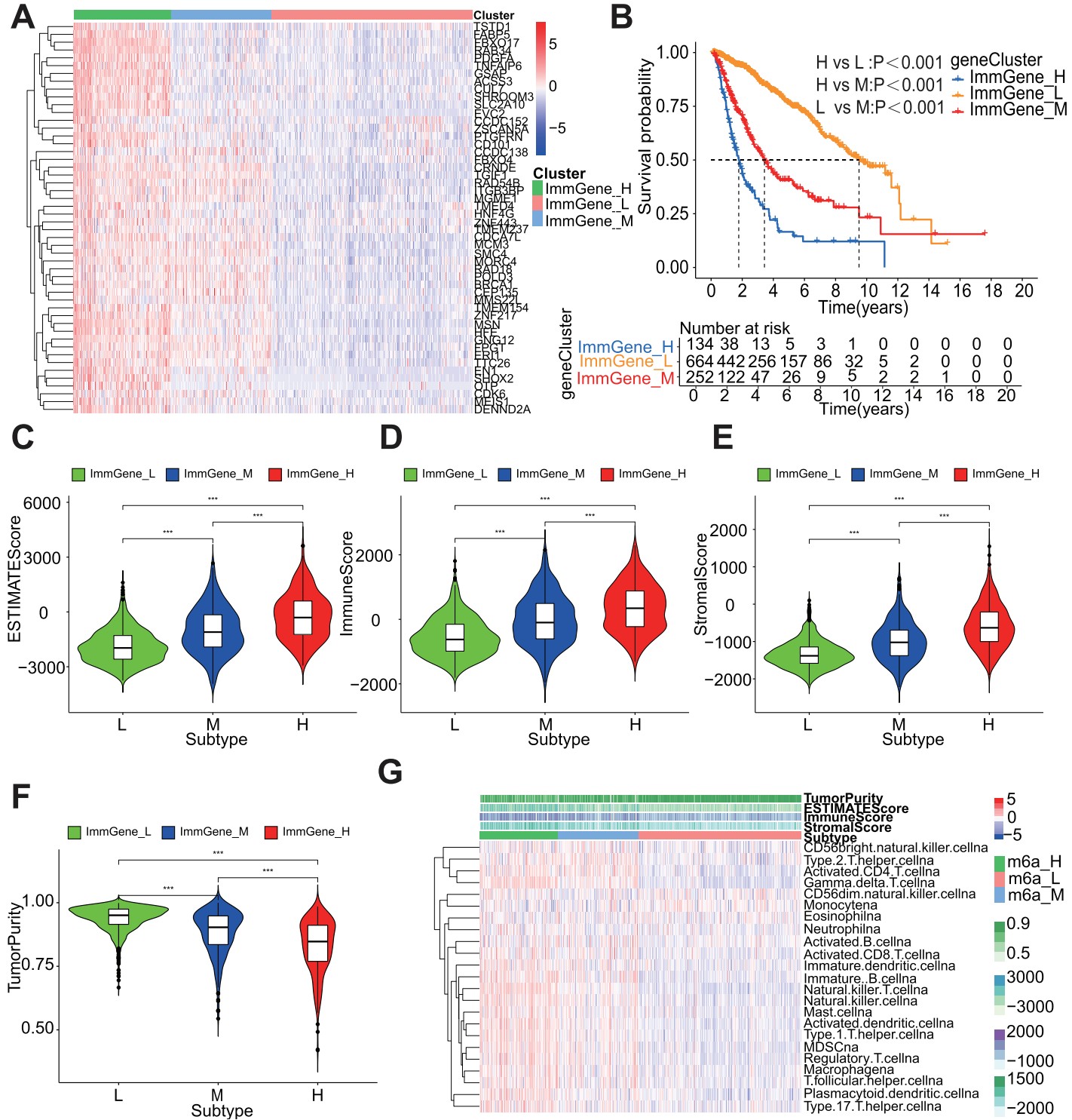

**Figure 5 Sample clustering based on prognostic m6A-related immune genes and tumor microenvironment (TME) scoring.** (A) Clustering heatmap. Displays the classification and expression levels of each gene. (B) Survival curves for the three different expression groups. (C) Violin plot of ESTIMATES scores across different groups. (D) Violin plot of immune scores across different groups. (E) Violin plot of stromal scores across different groups. (F) Violin plot of tumor purity across different groups. (G) Heatmap. The x-axis represents sample names, and the y-axis represents different immune cell types and their corresponding TME scores. In the violin plots, ***$P < 0.001$.

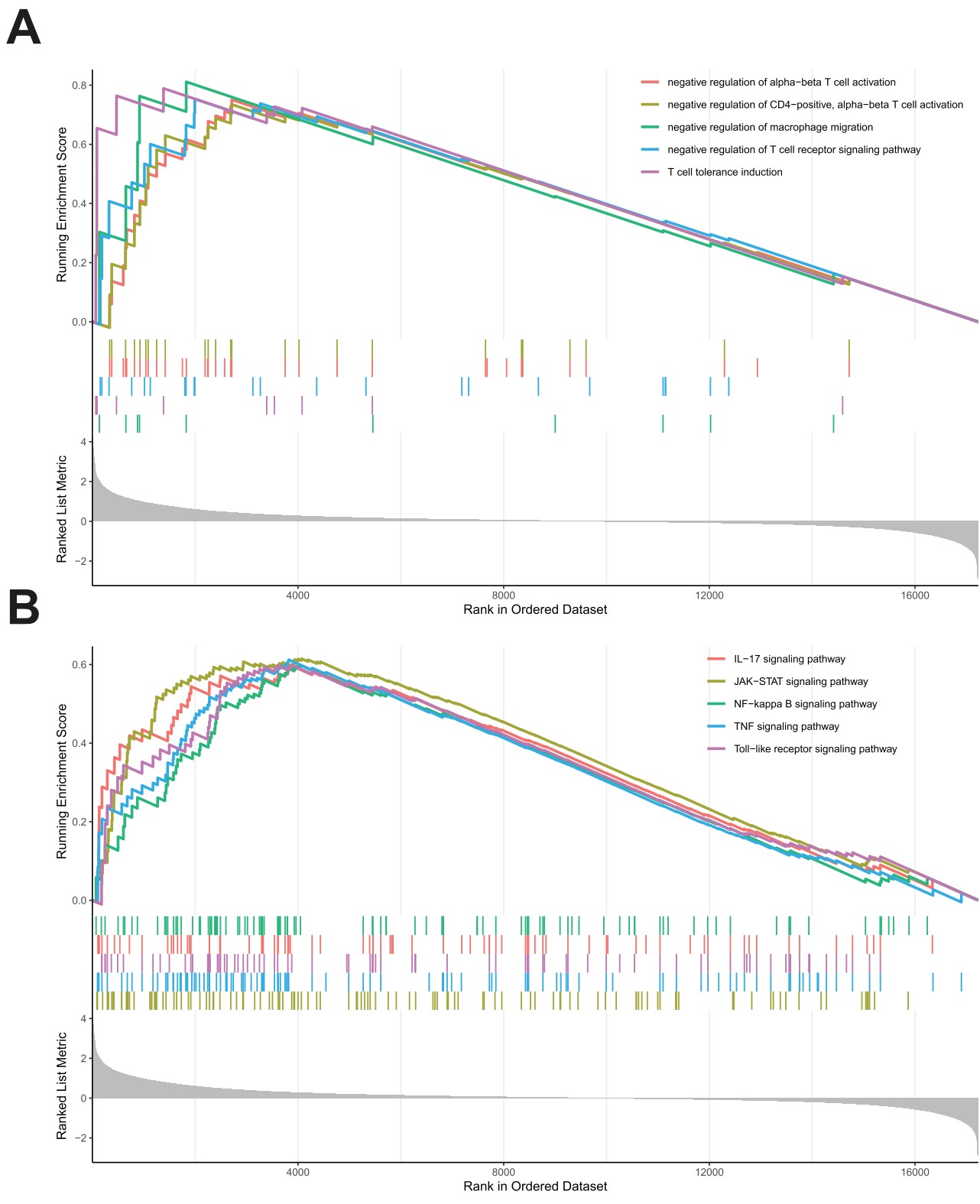

**Figure 6 GSEA enrichment analysis.** (A) GO enrichment results show GO terms that were significantly enriched in the immune gene high-expression group. (B) KEGG enrichment results show KEGG pathways that were significantly enriched in the immune gene high-expression group.

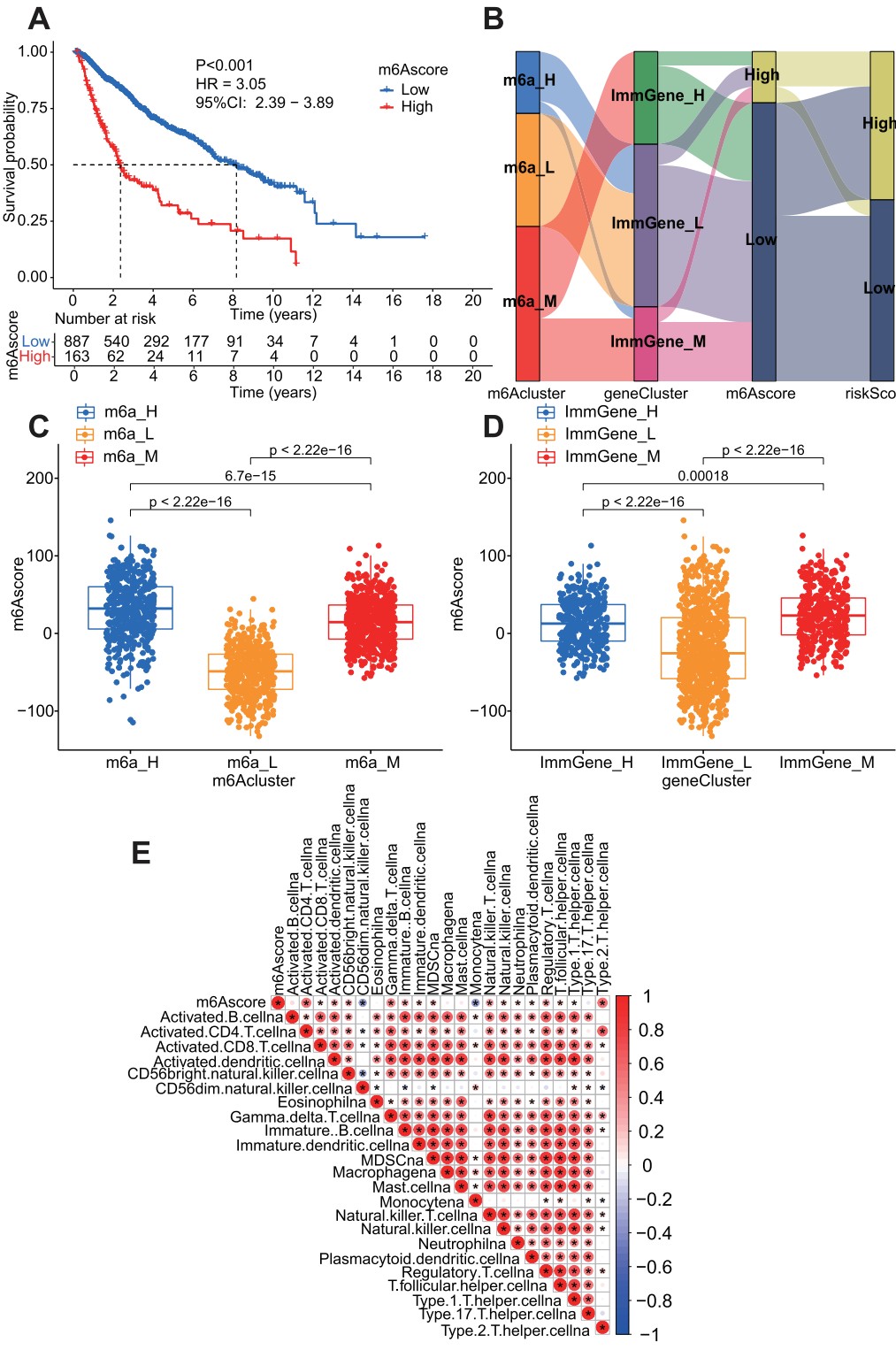

**Figure 7** **m6Ascore and two fractal types.** (A) m6Ascore survival analysis, the *P* value, hazard ratio (HR) and 95% confidence interval are shown. (B) Sankey diagram. Visualization of the correspondence between m6Ascore and the two typologies. (C) Box line plot showing a comparison of m6Ascores between m6A fractions. (D) Box line plot showing a comparison of m6Ascores between immune gene types. (E) Correlation matrix for m6Ascore and immune cells. Red represents positive correlations, and blue represents negative correlations. Asterisks indicate that the correlation is significant.                               

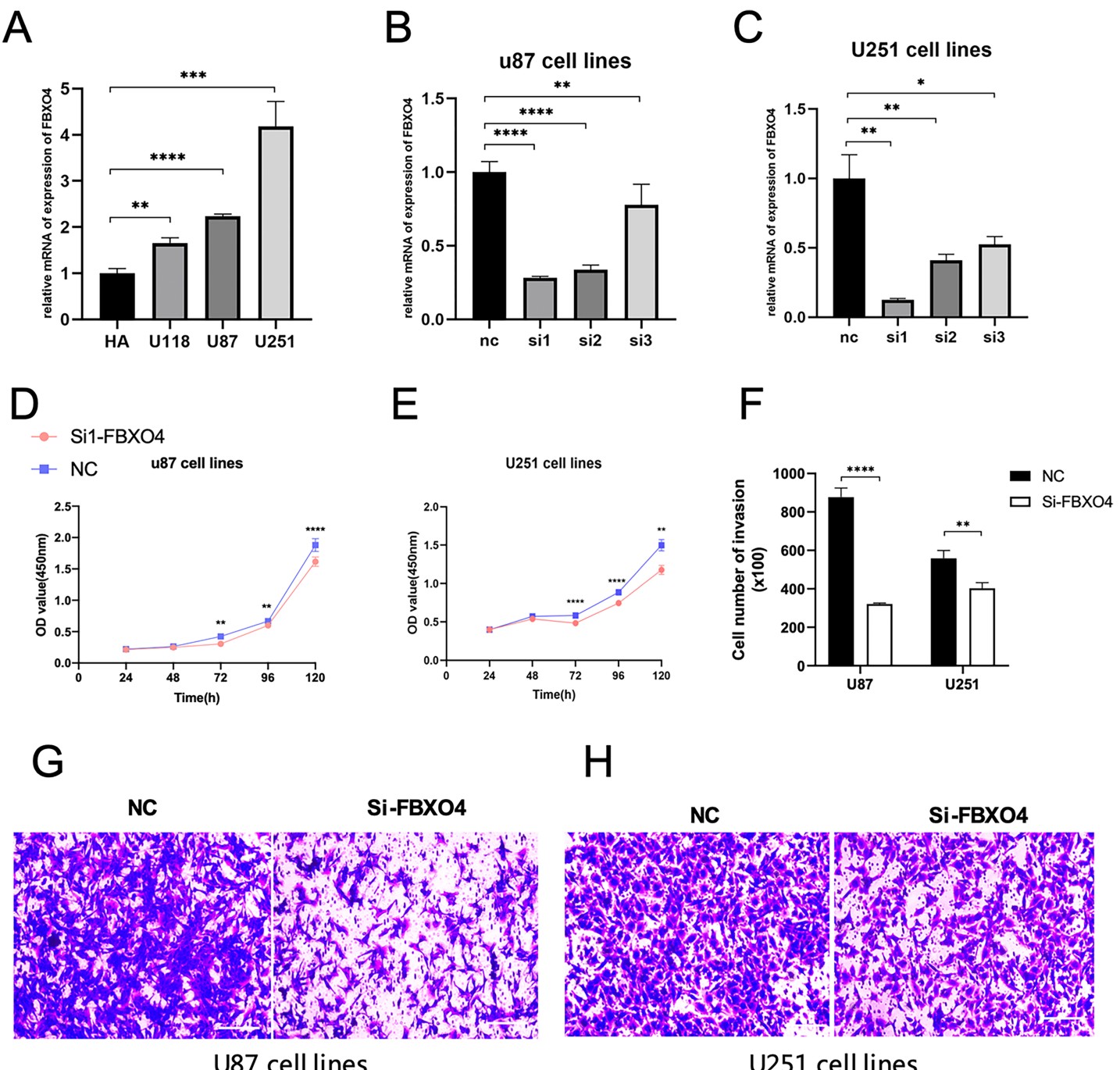

**Figure 8 Biological function of FBXO4 in glioma cells.** (A) The relative FBXO4 mRNA levels in glioma and HA1800 cells were measured by qRT-PCR. (B, C) qRT-PCR analysis confirmed that FBXO4 siRNA inhibited FBXO4 expression in U87 and U251 cells. NC-treated cells served as controls. (D, E) The CCK-8 test was used to find the validity of FBXO4 knockdown on the cell proliferation of U87 and U251 cells. (F) Quantification of migration cells. (G, H) Representative images of the transwell assay; the left panels show the U87 cell line, and the right panels show the U251 cell line. ****$P < 0.0001$, ***$P < 0.001$, **$P < 0.01$ *$P < 0.05$.

factors for LGG patients. The area under the 5-year ROC curve (AUC) values revealed that the efficacy of our prognostic model was significantly greater than that of the TIS and TIDE scores (Fig. S3).

## Downregulation of FBXO4 inhibits the proliferation and migration of glioma cell lines

The biological role of model genes in low-grade gliomas needs to be further investigated. Using the Gene Expression Profile Interaction Analysis (GEPIA) server (Fig. S3C), we identified a gene, FBXO4, that was independently associated with overall survival in low-grade glioma patients. Our results revealed that, at the RNA level glioma cell lines expressed higher levels of FBXO4 than did normal human astrocytic brain cells, especially the U87 and U251 cell lines (Fig. 8A). Next, we silenced the FBXO4 target gene in the U87 and U251 cell lines (Figs. 8B, 8C). In addition, the siRNA target sequences we used (Si1, Si2, and Si3) have been uploaded (File S7). Next, we uniformly used the identified relatively efficient gene silencing sequences, *i.e.*, si1, for the corresponding experimental studies. The CCK8 assay revealed that the proliferative capacity of glioma cells in the FBXO4-silenced group was significantly lower than that of the normal group (Figs. 8D, 8E). Next, in transwell assays, FBXO4 knockdown significantly inhibited the migration of U251 and U87 cells (Figs. 8F–8H). These results suggest that FBXO4 gene knockdown can inhibit the proliferation and migration ability of glioma cells.

## DISCUSSION

Low-grade gliomas (LGGs), although slow-growing, carry a high risk of recurrence and poor long-term prognosis (*Saijo et al., 2024*). In recent years, with the deepening of research on m6A proteins, the mechanisms of m6A in cancer have gradually been revealed. This study systematically explores the relationship between m6A regulatory genes and the immune microenvironment of LGG by constructing a prognostic model based on m6A-related immune genes. Our findings indicate that inhibiting m6A-related proteins not only reduces methylation-associated immune evasion but also decreases the renewal efficiency of tumor cells, providing new potential targets for individualized treatment of LGG (*Su et al., 2020*). Additionally, the high m6Ascore group is significantly associated with insufficient immune infiltration and reduced patient survival (*Zhang & Read, 2018*), a finding consistent with *Huang, Weng & Chen (2020)* in melanoma, where m6A protein knockout enhances sensitivity to interferon-gamma (IFNγ), thereby improving the efficacy of anti-PD-1 inhibitors (*Huang, Weng & Chen, 2020*).

Through multidimensional clustering analysis and ssGSEA, we further reveal that m6A-related phenotypes may promote the survival of immunosuppressive cells by increasing LGG stemness and stromal cell activation while inhibiting IFNγ, leading to treatment resistance and low responsiveness to immunotherapy (*Huang, Weng & Chen, 2020*). This finding aligns with *Yang et al. (2019)*, further supporting the critical role of m6A in the tumor immune microenvironment (*Yang et al., 2019*). Moreover, our prognostic model combines m6A immune genes and regulatory genes, enhancing its clinical translational value through dynamic nomograms. Compared to existing studies (*Gittleman, Sloan & Barnholtz-Sloan, 2020*; *Tan et al., 2020*), our model not only provides more accurate survival predictions but also, for the first time, links m6A-related genes to the immune characteristics of LGG.

Among the model genes, CRNDE, a long non-coding RNA, is upregulated in various solid tumors and promotes malignant progression of glioma by regulating the miR-384/PIWIL4/STAT3 axis (*Zheng et al., 2016*). Studies show that CRNDE expression regulates glioma cell growth and promotes tumor progression by positively regulating the EGFR signaling pathway (*Kiang et al., 2017*). Additionally, CRNDE exhibits oncogenic effects in prostate cancer, hepatocellular carcinoma, cervical cancer, and bladder cancer (*Liu et al., 2016*; *Tang, Zheng & Zhang, 2018*). Although CRNDE has been widely studied, its specific mechanisms in LGG require further exploration. CUL7, an E3 ubiquitin ligase, promotes glioma invasion by activating the NF-κB pathway (*Xu et al., 2020a*), and its high expression is significantly associated with poor patient prognosis (*Kong et al., 2019*). TNFAIP6, a secreted product of tumor necrosis factor-stimulated gene 6 (TSG-6), plays an important role in tumor prognosis (*Chan et al., 2019*), but its role in LGG needs further validation (*Lin et al., 2022a*).

Notably, FBXO4, a newly identified gene in this study, has not been extensively studied in LGG. FBXO4 is a key component of the Skp1-Cul1-F-box E3 ligase complex and is closely related to the development of various tumors (*Feng, Yang & Wang, 2017*; *Qie et al., 2017*). Our *in vitro* experiments show that interfering with FBXO4 expression inhibits the proliferation and migration of U251 and U87 glioma cells, suggesting its potential value in LGG diagnosis and prognosis. Although we have preliminarily validated the function of FBXO4, its specific mechanisms in LGG require further exploration through additional experimental data.

The significance of this study lies in its first-time linkage of m6A-related immune genes to the prognosis and immune microenvironment of LGG, providing new insights and tools for individualized treatment of LGG. By integrating clinical data and genomic databases, we can offer personalized treatment plans for different patients and more accurately predict disease progression (*Grinfeld et al., 2018*). Furthermore, this study reveals the complex interactions between m6A-related genes and the LGG immune microenvironment through multidimensional analysis, providing important directions for future research. However, this study has some limitations. First, due to database constraints, we were unable to include key clinical information such as tumor size, resection extent, comorbidities, and treatment regimens, which may potentially impact the interpretation of results. Second, although we preliminarily validated the functions of model genes through the GEPIA database and *in vitro* experiments, further research is needed to clarify their specific mechanisms. Additionally, the sample size of this study is limited, and larger-scale clinical data are required in the future to validate the accuracy and applicability of the model.

Future research should focus on further optimizing the predictive power of the prognostic model through multi-omics data and larger clinical samples, and delve deeper into the interaction mechanisms between m6A regulatory genes and the LGG immune microenvironment, particularly the specific role of FBXO4 in tumor immune responses. Moreover, validating the reliability of model genes through experimental data and assessing their potential as therapeutic targets will also be important directions for future research. Exploring the relationship between m6A-related genes and other tumor types to

expand the clinical application scope of this study will provide new insights into tumor immunotherapy.

In summary, this study provides new tools and insights for the individualized treatment of LGG by constructing a prognostic model based on m6A-related immune genes. Despite some limitations, our research fills the gap in the current literature regarding the relationship between m6A regulatory genes and the immune characteristics of LGG, laying an important foundation for future research and clinical practice.

## CONCLUSION

In brief, we have created a web-based dynamic nomogram tool and a prognostic model for LGG, both of which have the potential to be clinically translated. We have also learned more about the elements that affect the pathophysiology of LGG by examining the link between the two clustering models of LGG, the m6Ascore, and the elements of the tumor microenvironment. Additionally, we conducted a number of *in vitro* tests to confirm the importance of FBXO4, one of the model genes. As a result, the current study offers a useful method for precisely assessing the clinical prognosis of LGG patients as well as fresh ideas and inspiration for future studies into deeper mechanisms.

## ACKNOWLEDGEMENTS

The authors would like to thank the TCGA, CGGA and GSEA databases for the availability of the data. The authors would be like to thank the technical support by the Huazhong University of Science & Technology Analyttical & Testing center. We would like to Wenjun Zhu, Min Fu, Feng Yang, Ziqi Chen, Qiang Zhang, Bi Peng, Qianxia Li (Department of Oncology, Tongji Hospital, Tongji Medical College, Huazhong University of Science and Technology, Wuhan 430030, China) for their contribution.

## ABBREVIATIONS AND GLOSSARY

| | |
|---|---|
| **GSEA** | Gene Set Enrichment Analysis |
| **ROC** | Receiver operating characteristic |
| **KEGG** | Kyoto Encyclopedia of Genes and Genomes |
| **ssGSEA** | Single Sample Gene Set Enrichment Analysis |
| **GO** | Gene Ontology |
| **LASSO** | Least absolute shrinkage and selection operator |
| **TCGA** | The Cancer Genome Atlas |
| **CGGA** | The Chinese Glioma Genome Atlas |
| **AUC** | Area under the curve |

### Funding

This study was funded by the National Natural Sciences Foundation of China (Grant No. 82404195, 82003312, 82173311, 82303830). The funders had no role in study design, data collection and analysis, decision to publish, or preparation of the manuscript.

## Grant Disclosures

The following grant information was disclosed by the authors:
National Natural Sciences Foundation of China: 82404195, 82003312, 82173311, 82303830.

## Competing Interests

The authors declare that they have no competing interests.

## Author Contributions

- Yiling Zhang conceived and designed the experiments, performed the experiments, analyzed the data, prepared figures and/or tables, authored or reviewed drafts of the article, and approved the final draft.
- Na Luo conceived and designed the experiments, performed the experiments, analyzed the data, prepared figures and/or tables, authored or reviewed drafts of the article, and approved the final draft.
- Xiaoyu Li performed the experiments, analyzed the data, prepared figures and/or tables, authored or reviewed drafts of the article, and approved the final draft.
- Chuanfei Zeng analyzed the data, prepared figures and/or tables, authored or reviewed drafts of the article, and approved the final draft.
- Xin Chen performed the experiments, analyzed the data, prepared figures and/or tables, authored or reviewed drafts of the article, and approved the final draft.
- Xiaohong Peng performed the experiments, analyzed the data, prepared figures and/or tables, authored or reviewed drafts of the article, and approved the final draft.
- Yuanyuan Zhang conceived and designed the experiments, prepared figures and/or tables, authored or reviewed drafts of the article, and approved the final draft.
- Guangyuan Hu conceived and designed the experiments, prepared figures and/or tables, authored or reviewed drafts of the article, and approved the final draft.

## Data Availability

The Brain Lower Grade Glioma (phs000178) data is available at TCGA: https://portal.gdc.cancer.gov/projects/TCGA-LGG.

The raw data used in the analysis is available at FigShare: Zhang, Yiling; Luo, Na; Li, Xiaoyu; Zeng, Chuanfei; Chen, Xin; Peng, Xiaohong; et al. (2024). Supplemental File with TCGA raw data (TCGA clinical information and TCGA expression profile for LGG patients). figshare. Dataset. https://doi.org/10.6084/m9.figshare.27854115.v1.

The mRNAseq_693 and mRNAseq_325 datasets are available at CGGA: https://www.cgga.org.cn/download.jsp.

The codes used in this study are available at GitHub and Zenodo:
- https://github.com/Lilyyyyyy0207/FBXO4.git.
- Lilyyyyyy0207. (2025). FBXO4: V1 (Version V1). Zenodo. https://doi.org/10.5281/zenodo.14949758.

## Supplemental Information

Supplemental information for this article can be found online at http://dx.doi.org/10.7717/peerj.19194#supplemental-information.

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
