# Peer review of "The prognostic model of low-grade glioma based on m6A-associated immune genes and functional study of FBXO4 in the tumor microenvironment"

_PeerJ, doi:10.7717/peerj.19194_

## Round 0.1 · original submission · Major Revisions

Please address concerns of the revisers and revise manuscript accordingly.

Reviewer 1 ·

Basic reporting

Language and Clarity: The manuscript requires substantial improvement in English language usage. The current text contains numerous grammatical errors, unclear sentences, and a lack of professional, scientific tone. I recommend the authors seek help from a native English speaker or a professional editing service to enhance the manuscript's readability and ensure it meets the standards expected by an international audience. Too many lines are examples of challenging language to be enumerated here.
Introduction and Background: The introduction needs to provide more context to establish how this study fits into the broader field of research. The rationale for choosing m6A-associated immune genes in low-grade gliomas (LGG) context should be more precise. The authors should expand on the knowledge gap they intend to fill and provide more relevant background literature to justify their research objectives.
Structure and Data Presentation: The article currently lacks a clear and logical structure. Some sections are missing or not well defined, such as the aim of the study and a cohesive flow from one section to another. The titles of the results section should represent the findings and not the names of the methods used. Results sections shouldn’t contain conclusions such as, for example, row 258. Conclusions should be kept for the Conclusions part. Additionally, figure legends and descriptions are incomplete or unclear. Each figure should be clearly labelled with a comprehensive legend that allows it to be understood independently of the main text. All figure panels need to be mentioned individually in the results text and referred to as Figure 1 A, Figure 1B and so on. Besides, all figure panels mentioned must be described and adequately incorporated in the Results text with a logical flow in mind.

Experimental design

Research Question and Aim: The manuscript does not clearly define its research question or the specific aim of the study. It is essential to articulate a well-defined, relevant, and meaningful research question linked to an identified knowledge gap in the current literature.
Methodological Clarity: The methods section lacks sufficient detail to allow another investigator to replicate the study; some parts contain unintelligible sentences (e.g., row 199 ‘after reversal’), or the described method is impossible to understand due to improper language use.

Validity of the findings

Data Robustness and Statistical Soundness. The results are presented without a logical flow, and it is challenging to understand why each analysis was done and why in the given order. Due to the manuscript's lack of logical flow and poor language use, it is difficult to follow or understand if the methods used are the proper ones. Basic biology notion mistakes, such as in rows 250 and 251, ‘expression of m6A regulated genes, ’ make it unintelligible to which genes the authors are referring because there is a difference between genes that are regulated by m6A RNA methylation and genes that have a role in m6A RNA methylation regulation.
Discussion: The discussion should be limited to the results presented in this article. The Discussion contains data from the literature without putting the current article findings in the context of the literature.

Additional comments

Strengths: The manuscript addresses a novel area by focusing on m6A-associated immune genes in low-grade glioma, which is commendable. Multi-factor Cox regression and LASSO analyses are used to construct the prognostic model, which shows a rigorous approach.
Weaknesses: Major revisions are necessary in terms of language, clarity of the aim, logical flow of the manuscript, and completeness of data presentation. The research's contribution to filling the identified knowledge gap should be more clearly articulated.

·

Basic reporting

The manuscript is well-written, with clear English and relevant figures, providing valuable insights into m6A-associated immune genes in low-grade glioma. However, formatting issues, unclear figure details like missing axis labels and p-values, and the need for control groups limit the strength of its conclusions.

Experimental design

The manuscript addresses a relevant research question by exploring a prognostic model for low-grade glioma, contributing to the identified knowledge gap in m6A-associated immune genes. The investigation is technically sound, but certain aspects, such as the rationale for selecting FBXO4 and the lack of control groups, weaken the overall rigor. While the methods are generally described, some key details—particularly in the model-building process and m6Ascore calculation—lack sufficient clarity for full replication. Ethical standards appear to be met.

Validity of the findings

The findings in the manuscript are relevant but somewhat overstated, particularly regarding the model’s performance and FBXO4 as a biomarker. While the data provided are generally robust, the lack of critical controls and details in the statistical analyses weakens the conclusions. The conclusions are linked to the research question.

Additional comments

The manuscript presents a prognostic model for low-grade glioma (LGG) based on m6A-associated immune genes, with a particular focus on the identification of FBXO4 as a potential biomarker. The study offers valuable insights, but several key concerns need to be addressed. For example, the overstated model’s performance; the rationale for prioritizing FBXO4 over other genes is unclear; and crucial methodological details are missing, including the absence of appropriate control groups in multiple analyses. Additionally, important figure details, such as p-values and hazard ratios for Kaplan-Meier plots, are missing across the manuscript. Please see below for detailed comments.

In the abstract, the statement that the model 'worked very well and was able to be applied with an area under the curve of over 0.9' could be overstated. This high AUC likely reflects the performance on training data, which is expected to be better. However, the external validation AUC values (Figure 3) are much lower, closer to 0.7, which indicates a more moderate performance, typical for real-world data. Additionally, the claim of FBXO4 as a new biomarker is premature. Beyond in vitro experiments, it requires further validation before FBXO4 can be confidently labeled as a biomarker.

Figure 1
Figure 1C lacks axis labels and the sample size is not provided for the tumor and normal groups.
Table 1. The table provides a detailed breakdown of clinical characteristics (age, gender, grade, survival status, etc.). However, there are likely other confounding factors that could impact the prognostic model and are not included here, such as tumor size, extent of resection, comorbidities, and treatment regimens beyond chemotherapy and radiation therapy. These factors could also influence patient outcomes and, if not adjusted for, could bias the prognostic model.
Figure 2
1. The authors should provide a clear mapping of numbers to gene names, either in a legend or supplementary figure, to make the LASSO coefficient plot more interpretable.
2. While LASSO regression is used to reduce the number of genes, it is not fully clear how the authors arrived at these specific four genes. The selection criteria should be better explained. In addition, how these 3229 genes were selected in the first place from CGGA?

Figure 3
1. The inclusion of the training (Figure 3A) and internal validation group (Figure 3B) results is expected to show strong performance since the model was trained and validated on these datasets. However, the external validation sets (Figure 3C/D) are more important for assessing the model's generalizability. I recommend focusing on the external validation results in the main figure, as they provide a more accurate representation of the model's real-world predictive power.
2. The Kaplan-Meier plots in Figure 3 are missing the hazard ratio (HR) and corresponding confidence intervals, which are critical for understanding the magnitude of the survival difference between the high-risk and low-risk groups. The suggestion is also applicable to other figures with KM plots (Fig 5, 6, 8,11 etc)
3. An AUC of around 0.7 indicates moderate discriminative ability but is not exceptional. Without a comparison to simpler models—such as a Cox regression model using basic clinical features like age, gender, and tumor grade—it’s difficult to determine the added value of the gene-based risk score. Introducing a simpler benchmark model would clarify whether the gene-based model offers significant improvements over standard clinical predictors.
Figure 5
1. While it might seem intuitive to expect the low-risk group to have fewer mutations, the opposite is observed with 99% in the low-risk group versus 93.7% in the high-risk group. More importantly, this comparison is not particularly informative without considering the types of mutations involved. A more meaningful approach would be to focus on driver genes and deleterious mutations that are more likely to impact tumor progression and patient outcomes.
2. It is essential for the authors to clarify how the tumor mutational burden (TMB) was classified as low (TMB-L) and high (TMB-H) but the description is missing.

Figure 6-9
1. Upon reviewing the clustering (Figure 6A, 7A) and survival analysis (Figure 6B), there appears to be minimal distinction between the m6A-L and m6A-M groups. The survival outcomes for these two groups are nearly identical, while only the m6A-H group demonstrates a significant survival difference. The authors should provide statistical comparisons, such as p-values, between the m6A-L and m6A-M groups. Without significant differences between these groups, the rationale for dividing the samples into three clusters rather than two is unclear. A binary classification (e.g., high and low) may be more appropriate unless further justification for the three groups is provided.
2. Consider combining some of these figures (e.g., 6 and 7, or 8 and 9). It could improve readability and make the results easier to digest. The main takeaways regarding immune infiltration, tumor purity, and survival could be consolidated, especially since the overall conclusions from these plots are quite similar.

Figure 11
In Figure 11 C and D, there is no control group or baseline reference provided to show how the m6Ascore compares against normal tissues or baseline gene expression levels. Without a control, it’s unclear if the differences in m6Ascore between clusters (e.g., m6A_H, L, M) are specific to tumor biology or are simply reflective of natural gene expression variability in a broader population.

Figure S1-3
While the analyses presented in these figures demonstrate significant associations between the four genes (CRNDE, CUL7, FBXO4, TNFAIP6) and survival outcomes, as well as their correlations with clinical factors, immune scores, and tumor microenvironment characteristics, together with the previous comment for figure 11, I strongly suggest the inclusion of a control group in these analyses. It could be a "housekeeping gene" that is typically not associated with cancer progression or immune response.

Figure 12
The decision to prioritize FBXO4, based solely on the GEPIA database rather than as a LASSO-identified gene, undermines the connection to the earlier statistical analyses, weakening the rationale for its selection. The authors should have clarified earlier in the study why FBXO4 stands out from the other genes identified in their model.

Reviewer 3 ·

Basic reporting

Many sentences throughout the manuscript are unnecessarily too long. For example, line 29-30, “The prognostic model we studied for LGG worked very well and was able to be applied with an area
under the curve of over 0.9”—the authors should have written it as “the resulting prognostic model achieved an ROC-AUC of 0.9”.

There are many places where spaces between words and parentheses are missing, e.g., line 51, line 55, line 58, line 64, etc.

There are multiple places where citations are needed, e.g., line 52-53, line 93,

Line 112-113, “Therefore, the role of m6A has been widely recognized as a reliable biomarker” should change to “Therefore, m6A has been widely recognized as a reliable biomarker”.

The INTRODUCTION section seems a bit too long, I recommend the authors shorten it with more concise language. The last paragraph of the INTRODUCTION section should briefly include the findings.

Line 135, “which calculated a |correlation coefficient| > 0.4” sounds strange and I am not sure what it means.

Experimental design

Line 132, the GSEA website’s link and citation should be provided.

In general, tools/software used require citations, which are missing in many places, including but not limited to line 133 ( Limma), line 153 (maftools), line 155 (DyNom), line 162 (ConsensusClusterPlus).

Line 140, “as others have done”—it requires citations.

Line 127, “Data acquisition and differential analysis”—what does “differential analysis” mean? Does it mean differential gene expression analysis? In line 223, the term becomes “differences analysis”, which is very confusing and thus I urge the authors to standardize their terminologies and make them consistent.

Line 143-144, please explain what this sentence means: “We divide all the samples of LGG in the TCGA database into two groups, the training group, and the internal
validation group, while looking for the corresponding external validation group in the CGGA database.”

Line 150-152, “To determine whether the riskScore obtained from the model could be used as an independent prognostic factor, we conducted further univariate and multifactorial prognostic analyses on the training group (P<0.05)”. First, “riskScore” is not defined in advance. Second, what does “P<0.05” here mean? P-values are given after statistical tests are done, here it sounds like the authors pre-set a P-value threshold? This is a bit confusing.

Line 168-169, “Our results benefit from a significant enrichment of five Gene Ontology (GO) terms in the immune gene high expression group”—I’m not sure how does it mean that the result benefit from something. Do the authors mean they used these methods?

In Figure 1C, normal group’s number is much smaller than tumor group’s. The authors should justify this extreme data imbalance. Also, I don’t see where the number of normal group is mentioned.

Validity of the findings

In Figure 1A, the non-significant nodes should not be in the picture. Also, the gene names are hardly readable.

The text in all the figures are barely readable, either due to small size or due to low resolution.

In Figure 1B, the color of the central nodes are blocked by the gene names.

The visualization in this manuscript is not well organized. The whole point of figures in a scientific article is to transform complex results into simplified and readable messages for audience. However, this is not the case in most figures. For example, Figure 1C presents a gigantic heatmap containing all the overwhelming raw data where I’m not sure what are the take-aways. In contrast, the description for this figure in RESULTS section is very minimal, which is probably because there is nothing much to discuss on this result anyway. Same problem applies to Figure 6A, Figure 7A, Figure 8A, Figure 9A, and Figure 11E. Similarly, Figure 12G-H appears to be the staining results of different cell lines, but the legend does not explain what it is and thus the audience cannot tell what is going on here.

Most importantly, multiple comparison correction is missing throughout the methodology. So the validity of the findings remain elusive.

---

## Round 0.2 · Major Revisions

As you can see, two of the three original reviewers are still dissatisfied with the revision. Although one of the reviewers (R1) recommended rejection, I decided to give you another chance to reply to the critiques and revise you manuscript.

Reviewer 1 ·

Basic reporting

Language and Clarity: Although the authors have tried to improve the manuscript's language, it still lacks clarity and fails to maintain a professional, scientific tone. Several sentences remain ambiguous, and the overall presentation does not meet the journal's standards. It is recommended that the authors review other publications in PeerJ to better understand the expected level of scientific language and clarity.
Introduction and Background: While the introduction has been expanded, its quality remains inadequate. For example, rows 54-55 incorrectly state that low-grade glioma (LGG) includes grades II and III, despite the correct information in the cited article by Li et al. (2023). The introduction also fails to cite the authoritative WHO 2021 CNS tumour classification, which should be the primary reference for CNS tumour grading and classification. Furthermore, the rationale for focusing on m6A-associated immune genes in LGG is unclear, and the study's aims are not explicitly stated. The introduction is overly lengthy and does not effectively contextualise the study.
Structure and Data Presentation: The manuscript's structure is still suboptimal, lacking logical flow between sections. The selection criteria and methodology for identifying m6A regulators are not clearly explained. Graphical elements, such as Figure 1C, fail to provide meaningful information, as the figure lacks gene names, up-/down-regulation details, or contextual relevance. While some improvements have been made to figure legends and section titles, the results section remains fragmented, lacking a coherent narrative.

Experimental design

Research Question and Aim: The study still lacks a clear research question or specific aims, which undermines its scientific value.
Flaws in Experimental Design: A critical flaw lies in the data sets used for analysis. Table 1 includes grade III tumours, yet LGG comprises only grades I and II. Additionally, the data sets rely on outdated tumour classification criteria rather than the current WHO 2021 standards. A neuropathologist must accurately reclassify samples before proceeding with data analysis. Similarly, caution should be exercised with databases like GEPIA, as their classification systems may not align with current clinical standards.
Selection of FBXO4 Gene: The rationale for selecting FBXO4 for experimental validation is poorly justified. There is no clear connection between FBXO4 and the results presented in earlier sections. Furthermore, the experiments used high-grade tumour cell lines rather than LGG, rendering conclusions about FBXO4's role in LGG misleading. The abstract's statement that FBXO4's role was determined in LGG is particularly problematic, given that no LGG-specific cell lines were used.

Validity of the findings

Discussion: The discussion still does not discuss this manuscript's findings and does not place the current article's findings in the context of the literature.

Additional comments

The manuscript explores a unique area of research by investigating m6A-associated immune genes in LGG. However, the experimental design's flaws, outdated tumour classification, and unclear scientific narrative significantly limit its contributions.

·

Basic reporting

no comment

Experimental design

no comment

Validity of the findings

no comment

Additional comments

The authors addressed my concerns and comments in my initial review.

Reviewer 3 ·

Basic reporting

The authors have addressed most of my previous concerns on wordings, ambiguity, and text length.

Experimental design

The authors have improved the manuscript in terms of writings, logics, and citations. However, there is one point that remains an issue to me:

The extreme data imbalance--normal group (N=5) vs. tumor group (N=529).

I understand that it is difficult to obtain enough samples, but results derived from this extreme data imbalance could be problematic.

Validity of the findings

The authors have improved most of the figures in terms of clarity and organization.

However, the authors still did not undergo multiple comparison correction.

Multiple comparison correction is a standard statistical procedure while performing differential gene expression analysis to avoid false positive discovery. The authors should do it for a more rigorous study, or properly justify why they think it's not necessary.

---

## Round 0.3 · accepted · Accept

All remaining concerns of the reviewers were addressed, and the revised manuscript is acceptable now.

Reviewer 3 ·

Basic reporting

The authors have addressed all my comments.

Experimental design

The authors have addressed all my comments.

Validity of the findings

The authors have addressed all my comments.